# WHEN SUFFICIENCY IS INSUFFICIENT: PROBABILISTIC NEURAL REPRESENTATIONS AS AN INFORMATION BOTTLENECK

## ABSTRACT

The neural basis of probabilistic computations remains elusive, even amidst growing evidence that humans and other animals track their uncertainty. Recent work has proposed that probabilistic representations arise naturally in task-optimized neural networks trained without explicitly probabilistic inductive biases. However, prior work has lacked clear criteria for distinguishing probabilistic representations—those that perform transformations characteristic of probabilistic computation—from heuristic neural codes that merely reformat inputs. We propose a novel information bottleneck framework, the *functional information bottleneck (fIB)*, that crucially evaluates a neural representation based not only on its statistical sufficiency but also on its *minimality*, allowing us to disambiguate heuristic from probabilistic coding. To demonstrate the power of this framework, we study a variety of task-optimized neural networks that had been suggested to develop probabilistic representations in earlier work: networks trained to perform static inference tasks (such as cue combination and coordinate transformation) or dynamic state estimation tasks (Kalman filtering). In contrast to earlier claims, our minimality requirement reveals that probabilistic representations fail to emerge in these networks: they do not develop minimal codes of Bayesian posteriors in their hidden layer activities, and instead rely on heuristic input recoding. Therefore, it remains an open question under which conditions truly probabilistic representations emerge in neural networks. More generally, our work provides a stringent framework for identifying probabilistic neural codes. Thus, it lays the foundation for systematically examining whether, how, and which posteriors are represented in neural circuits during complex decision-making tasks.

## 1 INTRODUCTION

Under a particular generative model of the world that prescribes how latent variables generate observations, the brain may employ one of two broad classes of recognition models to estimate those latent variables (Koblinger et al., 2021): probabilistic models that employ Bayesian inference (MacKay, 2005), and non-probabilistic models that compute intermediate values that do not necessarily correspond to posteriors over latent variables, such as function approximating neural networks (LeCun et al., 2015). Probabilistic neural representations are advantageous over heuristic ones because they facilitate flexible modularity across computational circuits, information fusion under nuisance variation (such as during temporal filtering), optimal information gathering for maximally reducing one's uncertainty, and learning the structure of the world (Koblinger et al., 2021). From a machine learning perspective, they also correspond directly to the notion of adversarial robustness (Carlini & Wagner, 2017): networks that act according to a posterior are inherently invariant to nuisance perturbations. Although behavioral evidence suggests that human and other animals are uncertainty-aware in perceptual judgements (Ernst & Banks, 2002; Körding & Wolpert, 2004; Boundy-Singer et al., 2024), it remains unclear whether uncertainty is represented probabilistically—*i.e.,* that neural circuits themselves compute with probabilities—or heuristically via dedicated channels for processing uncertainty (Lippl et al., 2024; Rahnev et al., 2021) or by circumventing explicit uncertainty representation altogether LeCun et al. (2015).

Recent work has suggested that neural networks develop robust internal representations of posteriors even without explicit probabilistic inductive biases (Orhan & Ma, 2017), suggesting that probabilistic representation is an emergent property of near-optimal behavior. However, previous decoding approaches (Orhan & Ma, 2017; Walker et al., 2020) did not separate genuinely probabilistic representations—codes that support the characteristic transformations of probabilistic computation—from trivial reformatting of inputs. To resolve this ambiguity, we seek to identify the defining features that distinguish a probabilistic representation from a non-probabilistic one.

Recent debates in probabilistic coding have clarified that defining probabilistic representation ultimately depends on how we define representation itself (Rahnev et al., 2021; Lippl et al., 2024; Walker et al., 2023). Building on this view, we connect general criteria for representations to an information bottleneck perspective on probabilistic inference. From this perspective, neural activity constitutes a probabilistic code only when it is approximately sufficient and minimal—preserving exactly the information needed for behavior and generalization, and nothing more. Concretely, task-relevant posteriors should be decodable from hidden activities of networks that exploit uncertainty to behave optimally, but if raw inputs are also decodable, the network has not transformed them into a usable code for downstream computation. This reframes inference as compression, a long-standing idea in Bayesian (MacKay, 2005) and Minimum Description Length (Grünwald, 2007) approaches to machine learning but one that has rarely been applied to distinguish neural representations of uncertainty. Here, we demonstrate how this compression-based view can sharpen our understanding of probabilistic coding.

Classical information bottleneck approaches to deep learning and neural coding have been instrumental in understanding training dynamics (Shwartz-Ziv & Tishby, 2017), characterizing flat minima (Achille & Soatto, 2018), and unifying competing theories about sensory encoding (Chalk et al., 2018). However, these analyses are critically limited by their reliance on estimating mutual information, an intractable and sometimes vacuous quantity to estimate in neural networks (Saxe et al., 2019; Goldfeld et al., 2019; Goldfeld & Polyanskiy, 2020). To circumvent this limitation, we propose the *functional information bottleneck (fIB)* a framework wherein we measure information content by training linear and nonlinear decoders ("probes") on the hidden activations of fully trained networks (Alain & Bengio, 2018; Li et al., 2024). This not only offers a tractable alternative to mutual information but also provides a more principled test about the format of neural codes. Unlike mutual information, which only measures statistical dependence, probing reveals which information is organized such that it can be efficiently used by downstream neurons (Xu et al., 2020).

We study a variety of task-optimized neural networks that had previously been suggested to develop probabilistic representations (Orhan & Ma, 2017): networks trained to perform static inference tasks (such as cue combination and coordinate transformation) or dynamic state estimation tasks (Kalman filtering). While these tasks and their corresponding networks are relatively simple, they are precisely the settings where the relevant posterior is analytically known and experimental evidence suggests that humans and other animals perform probabilistic computation (Ernst & Banks, 2002; Wolpert & Ghahramani, 2000; Schlicht & Schrater, 2007; Wu et al., 2006), allowing us to test whether such representations are truly probabilistic. Using more complex datasets (e.g., images) would obscure the underlying computational question because the ground-truth posterior is unknown; thus, one would not be able to validate our fIB framework without a well-defined generative model. Therefore, we intentionally keep our generative models simple to illustrate the basic principle underlying the framework.

Using our novel approach, we demonstrate that, contrary to the findings of (Orhan & Ma, 2017), task-optimized neural networks do not generically form probabilistic representations.

## 2 PROBABILISTIC REPRESENTATION AS AN INFORMATION BOTTLENECK

For a function, $\mathbf{r} = f(\mathbf{X})$, to be a specific probabilistic representation, where $\mathbf{X}$ denotes the inputs to the function and $p_z = p(z|\mathbf{X})$ is some target posterior, we want $\mathbf{r}$ to sufficient for the posterior $p_z$, and we would like for it to be invariant to nuisances $\boldsymbol{\nu}$ (Achille & Soatto, 2018; Walker et al., 2023). For some generic measure of information content $I(\cdot)$, sufficiency is expressed as $I(\mathbf{r}; p_z) = I(\mathbf{X}; p_z)$, or that the representation $\mathbf{r}$ is maximally informative about $p_z$. Invariance enforces that $\mathbf{r}$ filters out nuisance variables $\boldsymbol{\nu}$, *i.e.,* $I(p_z; \boldsymbol{\nu}) \approx 0$.

Traditional tests for invariance typically isolate known nuisance dimensions (Walker et al., 2020) or evaluate out-of-distribution generalization under carefully designed generative models (Orhan & Ma, 2017). While compelling, these approaches cannot exhaust the entire space of possible nuisances nor can they diagnose in-distribution redundancy. From a Bayesian perspective, compression arises not only from marginalizing over nuisance variables but also from eliminating irrelevant structure in $\mathbf{X}$ that does not affect $p_z$. Such compression supports robustness, enables flexible reuse of circuits across inference contexts, and reduces the computational burden on downstream decoders.

A more direct and stringent test for invariance is minimality (optimal compression). A code that is minimal with respect to input information but sufficient for representing the task-relevant posterior is necessarily invariant to all nuisance variation (Achille & Soatto, 2018). This yields an information-bottleneck-style criterion for probabilistic representation:

$$\mathbf{r}^\star = \operatorname*{arg\,min}_{\mathbf{r}:I(z;\mathbf{X}|\mathbf{r})\leq\alpha} I(\mathbf{X};\mathbf{r}),$$

The importance of verifying minimality is highlighted in the transfer learning experiments of (Orhan & Ma, 2017). A central advantage of probabilistic representations is their potential for flexible reuse across tasks. Koblinger et al. (2021) For example, an agent trained to estimate the latent stimulus $z$ driving two cues $\mathbf{X} = (\mathbf{x}_1, \mathbf{x}_2)$ should be able to reuse its internal representation if it needs to perform such cue combination with access to a third cue $\mathbf{x}_3$. This is implicitly a test for whether a representation of $\mathbf{X}$ is sufficient for representing $p_z$. Indeed, Orhan & Ma (2017) shows that when a network trained on $\mathbf{X}$ to estimate $z$ is frozen and its hidden representation $\mathbf{r}_{\text{perf}}$ is grafted onto another network with a third cue $\mathbf{x}_3$, this modular network optimally performs three-cue combination (Figure 1A). However, the same three-cue network performs equally well if $\mathbf{r}_{\text{perf}}$ simply encodes $\mathbf{X}$ itself (Figure 1B). This is a classic issue we refer to as the *readout fallacy*: a population may encode enough information to support a Bayesian decoder even if that population is not explicitly performing Bayesian computation. Put differently, $\mathbf{X}$ itself is trivially a sufficient statistics for $p_z$. Thus, only testing whether the task-relevant posterior is decodable from $\mathbf{r}_{\text{perf}}$ is insufficient in assessing probabilistic representation because a sufficiently expressive decoder can trivially decode the optimal posterior if $\mathbf{r}_{\text{perf}}$ encodes $\mathbf{X}$ instead of $p(z|\mathbf{X})$. Instead, if a specific neural population supports probabilistic computation, it must also compress away spurious correlations in $\mathbf{X}$ that would otherwise interfere with downstream inference and flexible reuse.

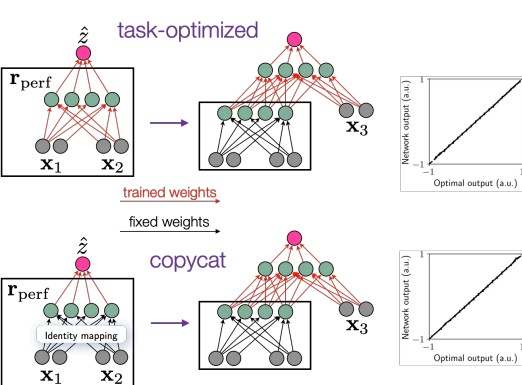

Figure 1: **Extending the task transfer setting from Orhan & Ma (2017) illustrates that testing sufficiency alone is not enough for identifying probabilistic representations. A.** Statistical sufficiency can be tested by training a module on two-cue combination and grafting it into a larger network with a third cue. If the extended network performs three-cue combination optimally without retraining, the module implicitly represents posterior uncertainty. **B.** However, equivalent performance may arise from a "copycat" network that merely passes inputs through, motivating a test of *minimal* sufficiency.

Importantly, minimality is only a requirement of a specific circuit or neural subpopulation, not the brain itself or even entire pathways. Notions of minimality are trivially violated when one considers photoreceptors as part of the same computational unit as downstream areas like V4, since early sensory areas are normatively viewed as input-preserving and, therefore, non-minimal. In fact, we would argue that decoding task-relevant posteriors from early sensory areas exactly illustrates the readout fallacy from above. Our goal is instead to identify which local circuits (e.g., subpopulations in V4/MT) contain representations sufficient for representing task-relevant posteriors and minimal with respect to their inputs. We will, therefore, refer to candidate representations that meet our criteria as *specific* probabilistic representations.

## 2.1 "Strong" specific probabilistic representation requires representational compression

Probabilistic computations, especially in the context of Bayesian perception and action, are many-to-one mappings. Therefore, if a representation $\mathbf{r}$ performs a probabilistic computation, it must throw away information by definition. We refer to this process as *representational compression*, by which a neural representation irreversibly compresses away task-irrelevant and nuisance variation. A neural representation that encodes the relevant posterior *and* exhibits representational compression constitutes a *strong* probabilistic representation: downstream circuits have access only to the task-relevant posterior and cannot recover task-irrelevant information. For instance, a strong probabilistic representation could assume a form $\mathbf{r} = p_z + \epsilon, \ \epsilon \sim \mathcal{N}(0, 1)$.

## 2.2 "Weak" specific probabilistic representation requires readout compression

In many settings, however, it may not be optimal to irreversibly destroy task-irrelevant information, especially when an agent is uncertain about which information will be relevant in future tasks. A population may therefore encode both the posterior and additional input features, provided that the posterior is easily accessible from a linear subspace. We refer to this as readout compression: irrelevant information is not deleted but is instead projected into the nullspace of the posterior subspace by an appropriate linear readout. Representations satisfying this criterion constitute weak probabilistic representations: these representations do not explicitly perform probabilistic computations themselves but facilitate computation downstream. For instance, applying some linear projection $R(\cdot)$ to both the inputs $\mathbf{x}$ and the posterior $p_z$—i.e., $\mathbf{r} = R(\mathbf{X}, p_z)$—would constitute a weak probabilistic representation because $\mathbf{r}$ contains a *linear subspace* that is minimally sufficient for $p_z$, even if irrelevant details in $\mathbf{X}$ are not globally compressed away.

## 2.3 Distinctions from previous work

It is necessary to point out a few key distinctions between our framework and the classical information bottleneck literature (Tishby et al., 2000; Slonim, 2002; Chechik et al., 2005). First, we do not assume $\mathbf{r}$ is a stochastic encoding of $\mathbf{X}$ (Strouse & Schwab, 2016). Two, we do not explicitly train any of our task-optimized networks with an IB objective, as has been proposed in (Alemi et al., 2019), so we do not choose $\alpha$; our analyses are all post-hoc. Third, we do *not* estimate mutual information directly, given its aforementioned complications (Goldfeld et al., 2019; Goldfeld & Polyanskiy, 2020). Instead, as we will discuss, we rely on probing decoders to approximate information content in the hidden layers of task-optimized networks.

## 3 Methods

Our approach requires training two types of neural networks: "performers" and probes. Performers are recognition models trained to perform a particular inference task. Probes are trained on the hidden activations of the performers to evaluate whether their internal representations are probabilistic.

### 3.1 "Performer" networks

#### 3.1.1 Static inference: cue combination and coordinate transformation

Following (Orhan & Ma, 2017), we trained feedforward "performer" networks to optimally perform cue combination and coordinate transformation; these networks are referred to as *task-optimized performers*. In both tasks, the inputs to the performer network consisted of two neural populations $\mathbf{x}_1, \mathbf{x}_2$, each with 50 independent Poisson neurons that had Gaussian tuning curves. The height of the neural population responses was modulated by a gain $\nu_i$, which was population-dependent and varied trial-by-trial. The activities of the input populations constituted the observations ("cues") based on which the performer networks needed to compute their outputs. In cue combination, both populations were driven by the same latent stimulus $z$, which the network had to estimate based on input layer activities (Figure 2A). In coordinate transformation, each population was driven by a different $z_i$, and the performer had to optimally estimate the sum of the two latent stimuli $z = z_1 + z_2$ (Figure 2B)—a more difficult task considering the network must marginalize out $z_1$ and $z_2$ (Ma et al.,

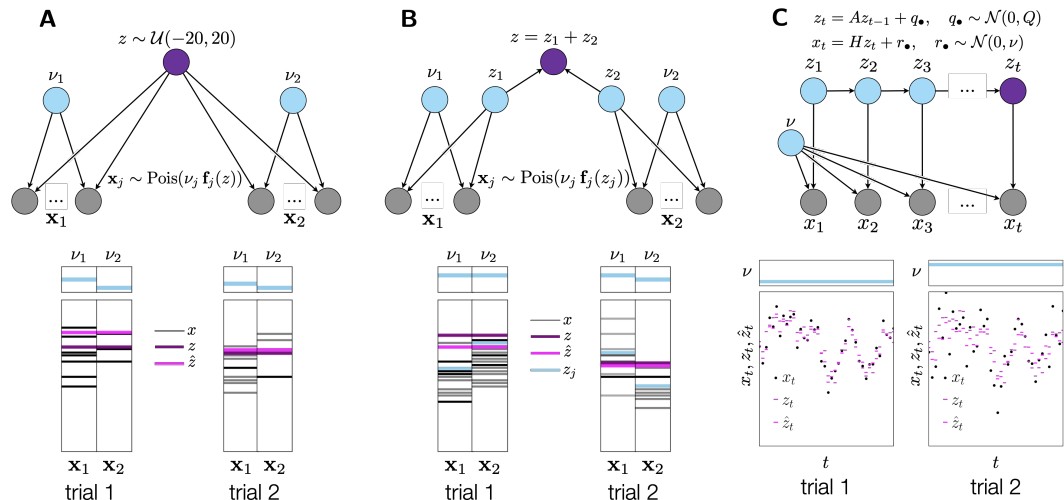

Figure 2: **Generative models used to train performer networks. A.** Cue combination: a single latent variable $z$ drives two independent neural populations modulated by nuisance variables $\nu_i$ (cf. Ma et al. (2006), Orhan & Ma (2017)). As shown in the two representative trials, gains are modulated from trial-to-trial and are population dependent. **B.** Coordinate transformation: two independent latent variables $z_1, z_2$ drive independent neural populations, and the inference task is to estimate the sum $z = z_1 + z_2$ (cf. Beck et al. (2007)). Again, nuisance variables $\nu_i$ are population dependent. **C.** Linear-Gaussian Kalman filtering: At each time point $t$, the performer must infer the value of the latent state $z_t$ given all previous observations $x_{1:t}$. The history of latent states $z_1, z_2, ..., z_{t-1}$ become nuisance variables on trial $t$. However, marginalizing over them is greatly simplified by 1) using recursive filtering and 2) constraining the system to be linear-Gaussian. We treat observation noise variance $\nu$ as a nuisance variable and compute the marginal filtering posterior over $\nu$.

2006; Beck et al., 2011). The gains $\nu_i$ for each input population were considered nuisance variables akin to psychophysical variables like contrast that the performers needed to marginalize out.

### 3.1.2 DYNAMIC INFERENCE: KALMAN FILTERING

To study probabilistic representations in dynamic inference tasks, we trained *recurrent* performer networks on the simplest form of dynamic state estimation: a 1-D linear dynamical system defined by the equations shown in Figure 2C. Here, $x_t$ and $z_t$ denote the observation and latent state at time $t$, respectively. $Q$ denotes the *process noise variance*, which directly modulates the true state, whereas $\nu$ is the *measurement noise variance*, which does not.

The well-characterized solution to estimating the latent state $z_t$ given a history of observations $x_{1:t}$ (when everything is linear-Gaussian) is the Kalman filter, which updates its state estimate by weighing incoming observations against previous state estimates based on their respective reliabilities. The higher the measurement noise $\nu$, the more the Kalman filter favors its previous state estimates, whereas higher process noise $Q$ means the Kalman filter will trust incoming observations more and make larger updates. Thus, we were interested in evaluating whether recurrent performers trained to perform Kalman filtering would implicitly understand how to weight incoming measurements based on their relative uncertainties. This is especially interesting under resource-constrained conditions where the number of hidden neurons is an order of magnitude smaller than the total number of observations (or total number of time steps $T$) because the network does not have enough capacity to memorize the entire sequence trajectory.

The results presented herein fix $A = 0.75, H = 1.0, Q = 0.5$, and $T = 40$. To mimic the nuisance "gains" from the PPC-style static inference tasks, we modulated measurement noise $\nu \in \{0.25, 0.5, 0.75, 1.0, 1.25\}$ from trial-to-trial (and fixed it within a trial). Because the performers were not given explicit information about $\nu$ from trial-to-trial, we consider the *marginal* Kalman filtering posterior whenever we performed our fIB analysis—that is, the posterior $p(z_t|x_{1:t}, \nu)$ marginalized over $\nu$. Therefore, posterior uncertainty reflects not just uncertainty about the state

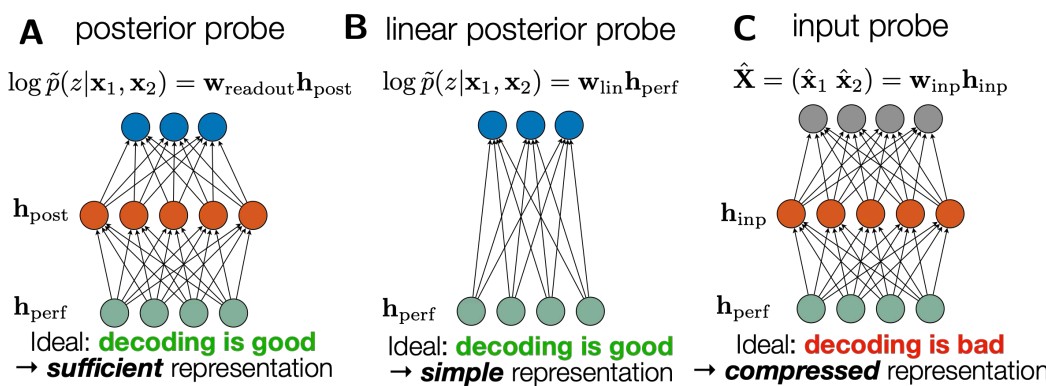

Figure 3: **Evaluating probabilistic representations via the fIB. A.** We use a nonlinear, high-capacity posterior probe to assess the sufficiency of a performer's internal representation $\mathbf{r}$. **B.** Using a linear posterior probe, trained with the same objective as the full posterior probe, we analyze whether internal representations of posteriors are in a simple and usable format. **C.** Finally, we use a nonlinear, high-capacity input probe to assess the degree of compression in $\mathbf{r}$ about inputs $\mathbf{X}$. Crucially, a probabilistic representation must satisfy poor input probe performance (compression) in addition to high posterior decodability (sufficiency).

estimate, which in Kalman filtering is monotonic and independent of the measurements $x_{1:t}$, but also uncertainty in $\nu$ itself, which makes the optimal posterior variance a function of the measurements and, thus, causes it to be temporally non-monotonic (see Appendix for more details).

## 3.2 TESTING GENERALIZATION

During training and testing, we varied the degree of "Bayesian transfer" (Orhan & Ma, 2017). This tested the networks' ability to generalize to unseen nuisance conditions. In the "all nuisances" condition, static inference networks were trained and tested on all pairwise gain combinations $(\nu_1, \nu_2)$ with $\nu_1, \nu_2 \in \mathcal{V} = \{0.25, 0.5, 0.75, 1, 1.25\}$, as in (Orhan & Ma, 2017). For the "interpolation" and "extrapolation" conditions, networks were trained on a subset of $\mathcal{V}$ and tested on the remainder of the set: $\nu_1, \nu_2 \in \{0.25, 1.25\}$ and $\nu_1, \nu_2 \in \{0.25, 0.5\}$, respectively. For Kalman filtering, the network was not explicitly given feedback about trial-to-trial measurement noise $\nu$. Therefore, much like the gains in the static inference tasks, the recurrent performer networks need to marginalize over $\nu$ to compute the marginal posterior $p(z_t|x_{1:t})$. Thus, in the "all nuisances" condition, networks were trained and tested on all possible measurement noise settings $\nu \in \mathcal{V} = \{0.25, 0.5, 0.75, 1, 1.25\}$. For the "interpolation" and "extrapolation" conditions, networks were trained using a similar $\nu$ schedule as in the static inference tasks, *i.e.,* using the appropriate subsets of $\mathcal{V}$.

## 3.3 APPROXIMATING INFORMATION CONTENT VIA THE FIB FRAMEWORK

We trained (also with Adam) two types of "probe" networks, posterior probes and input decoders, to determine whether task-optimized performers developed strong probabilistic representations (Figure 3). Posterior probes were multilayer ReLU networks (512 hidden neurons) trained (with a KL divergence loss) to decode a discretized version of the ground-truth posterior distribution[1] (which was analytically computable from input layer activations) from the hidden layer activations of the task-optimized networks. We chose to decode discretized posteriors—rather than the posterior sufficient statistics—because discretization enforces that the probe captures the entire posterior shape rather than just low-order moments. Although discretization is lossy, it constrained the probe to output valid probability distributions rather than arbitrary numbers (as would be the case if regressing sufficient statistics), which prevented trivial solutions and stabilized training. We trained a linear posterior probe with the same objective as the nonlinear posterior probe to assess weak probabilistic

---

[1]The posterior was quantized into 210, 209, and 300 bins (or neurons) for cue combination, coordinate transformation, and Kalman filtering, respectively.

representation. This also evaluated whether these networks behaved like probabilistic population codes (Ma et al. (2006)), which predict neural activity represents log-posteriors linearly.

Finally, we trained two kinds of input decoders for static versus dynamic inference tasks. For cue combination and coordinate transformation, the input probe was a two-layer ReLU networks with 200 hidden neurons trained to reconstruct the full input layer activations from its hidden layer, and these probes were trained with Poisson negative log-likelihood loss.

Given the temporal nature of dynamic inference tasks, decoding the entire input sequence—*i.e.,* a dynamical system trajectory—at each time step would not be appropriate. Instead, we trained a set of probes, each tasked with decoding a *fixed* lag in the sequence. For example, the lag 0 probe was, at every time step in the sequence, trained to predict the most recent measurement. We used lags $l = 0, 1, 2, 4,$ and 7, which meant that input probe training began only after a burn-in of 7 time steps in each trial; without this, comparing the lag 0 and lag 7 probes would be unfair, as the former would see more measurements. As these probes were each trying to decode only the scalar raw measurement at lag $l$, they were trained using mean squared error loss on the true measurement at lag $l$.

Nonlinear input and posterior probes were intentionally designed to be high-capacity decoders to minimize interactions between decoder structure and measured probe performance. However, in future work, we plan to explore the interaction between decoder capacity and probe performance. For example, if only very large capacity probes are required for reconstructing inputs, then perhaps simpler generic circuits may be functionally invariant to that information.

### 3.4 BASELINE PERFORMERS

To compare information content in the internal representations of task-optimized performers, we selected two suitable baseline performers. The first was a copycat network, which trivially copied inputs to its hidden layer. As illustrated previously, such a network is sufficient but almost never minimal[2], making it a natural lower bound on compression. For the static inference tasks, the copycat's input-to-hidden weight matrix was the identity matrix (with zero-padding) and only the hidden-to-output weights were trained. Extending this idea to Kalman filtering required additional care because of the temporal structure of the inputs. If the copycat were to present the entire input sequence to its hidden layer at every time step, then the hidden activity would remain identical across time within a trial, making the probing analysis ill-posed. Another alternative is for the copycat to represent only the observations revealed up to time $t$, so that hidden activity evolves with time. However, this raises the complication of appropriately handling the yet-unobserved future measurements—while zero-padding may be a natural solution, zero is itself a valid measurement value, so it conflates "unobserved" with "observed = 0," potentially leaking spurious information into the hidden representation. Hence, we instead designed a capacity-restricted copycat network as a sliding window buffer of size $w$ that was an order of magnitude smaller than the total number of time steps $T$. Since the task-optimized RNN cannot store unbounded history, a full copycat would give an inappropriate upper bound, since its memory capacity would exceed that of the RNN. Therefore, a sliding-window copycat approximates the best possible copycat heuristic under identical capacity constraints. Window size was chosen by finding the minimum $w$ for which a sliding-window Kalman filter—a Kalman filter with limited memory horizon—of length $w$ approximated the full Kalman filter up to a mean squared error cutoff of $10^{-9}$ (Supplementary figure 8). When training fIB probes, the first $w$ time steps of each trial were burnt in.

### 3.5 ADDITIONAL TRAINING DETAILS

All task-optimized performers were trained with mean squared error loss and stochastic gradient descent (Adam optimizer) and were trained for 10,001 epochs, which was sufficient for training loss to plateau. Feedforward (static inference) performers contained a single hidden layer with 200 neurons (and ReLU activations) and a final linear readout neuron. Recurrent (dynamic inference) performers consisted of a single gated recurrent unit (GRU) hidden layer, and a final linear readout neuron (since we consider only a 1-D filtering case). The number of hidden neurons in the recurrent layer ($H = 13$) was selected to match the window size of the copycat benchmark network (cf. Section 3.4).

---

[2]This is especially true in inference tasks constrained to the exponential family of distributions, where minimal sufficient statistics always exist.

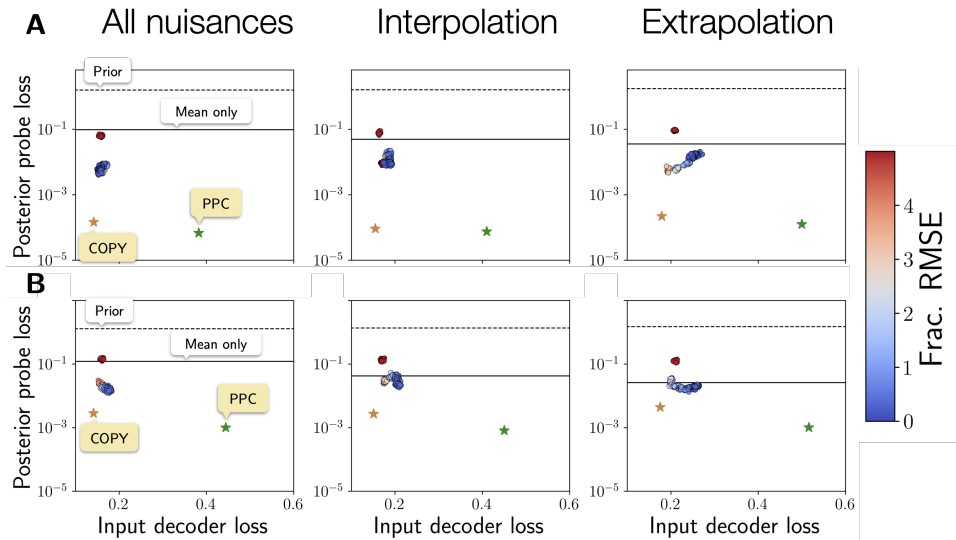

Figure 4: **fIB reveals that static inference performers are not minimal. A.** To evaluate networks using the fIB, we plot the nonlinear posterior probe loss (a proxy for sufficiency) against the input probe loss (a proxy for minimality), a visualization approach similar to the Information Plane from Shwartz-Ziv & Tishby (2017). Learning trajectories for cue combination performers are displayed as scatter plots colored according to the network performance at that stage of learning (in fractional RMSE compared to a Bayes-optimal model). Our two benchmark models, the copycat (COPY) and probabilistic model (PPC), are shown as orange and green stars, respectively. We show two horizontal calibration curves, the top (dotted) representing the performance of the prior and the bottom (solid) representing the performance of a "mean-oracle" network that does not explicitly encode trial-to-trial variability but instead learns to shift a posterior of *fixed* width to match the posterior mean (see Appendix for derivation). **B.** The same is shown for coordinate transformation. Columns correspond to the different Bayesian transfer settings. Note that the prior $p(z)$ for cue combination is uniform, whereas for coordinate transformation, the prior is the sum of two uniformly distributed variables $z = z_1 + z_2$, so $p(z)$ is, therefore, triangular.

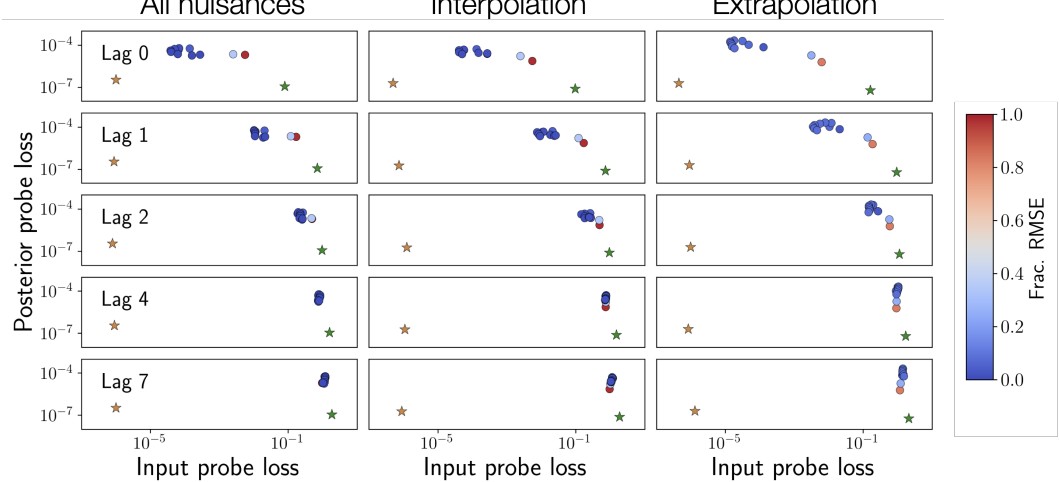

Figure 5: **RNNs trained on Kalman filtering also do not develop probabilistic representations.** As in Figure 4, learning trajectories for the RNN performers are displayed as scatter plots colored according to network performance at each stage of learning (in fractional RMSE compared to an optimal (marginal) Kalman filter). Rows correspond to increasing lags: a single input probe was trained to output one of the five fixed lags ($l = 0, 1, 2, 4, 7$) during each trial. Both benchmark performers are shown in the same format as in Figure 4.

## 4    RESULTS

### 4.1    TASK-OPTIMIZED NETWORKS ARE NOT STRONGLY PROBABILISTIC

The fIB framework reveals that networks perform a computation akin to input re-representation rather than explicit probabilistic representation (Figure 4). As expected, the probabilistic benchmarks (PPC) exhibit high posterior decodability while compressing their inputs away, whereas the copycat networks (COPY) trivially achieve high posterior decodability at the cost of compression, since they always have access to all inputs. The task-optimized performers in static tasks achieve satisfactory posterior decodability but crucially do so without compressing away input information. Networks remain mostly in the input re-representation regime where inputs are no more compressed than in the copycat networks but have clearly been reformatted to achieve satisfactory task performance. This result is consistent across generalization conditions and across tasks. Interestingly, Bayesian extrapolation (especially in cue combination) reveals a learning trajectory in which the network first attains high posterior decodability, then gradually compresses its inputs in a *lossy* manner. By the end of training, the network behaves like a "mean-oracle" network (solid horizontal line), which optimally estimates the posterior mean but erroneously assumes fixed trial-to-trial variability.

Surprisingly, contrary to prior empirical *and* theoretical work (Orhan & Ma, 2017; Koblinger et al., 2021), recurrent performer networks also fail to develop probabilistic internal representations, despite performing state estimation accurately under capacity constraints ($H < T$) and nuisance generalization (Supplementary figure 7). Although task-optimized performers are able to compress away input information, especially further back in the sequence history (Figure 5), the extent of this compression does not approach that of the optimal Kalman filter. Apparent parity with the PPC at lag $l = 7$ in Figure 5 is misleading, as the log scale of the abscissae conceals substantial differences in retained input information. Posterior decodability is considerably worse than both benchmark performers and, importantly, both posterior decodability (sufficiency) and input compression (minimality) *degrade* during training (Figure 5). This pattern persists across nuisance conditions and, unexpectedly, even when the hidden layer is reduced to $H = 5$

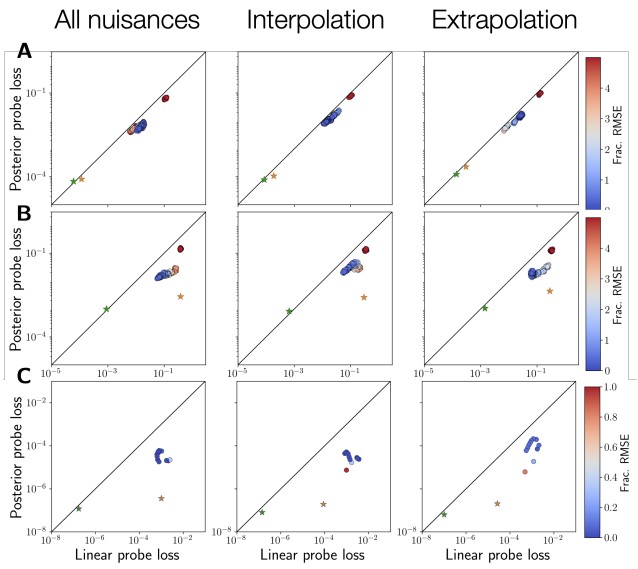

Figure 6: **Task-optimized performers do not form strongly linear representations in general.** Posterior and linear posterior probe loss are plotted against each other for **A.** cue combination, **B.** coordinate transformation, and **C.** Kalman filtering. Scatter plot and benchmark performer colors are the same as in Figures 4 and 5. The solid line indicates unity of linear and nonlinear posterior probe performance, signifying a simple linear internal representation.

neurons (Supplementary figure 10). In principle, such severe capacity limits should force the network to compress inputs into minimal sufficient statistics, since input recoding is explicitly suboptimal: the sliding-window Kalman filter can no longer encode enough inputs to reconstruct the full posterior. Even under these constraints, the networks perform well without encoding uncertainty more effectively than the copycat.

### 4.2    WEAK PROBABILISTIC REPRESENTATIONS FAIL TO CONSISTENTLY EMERGE IN TASK-OPTIMIZED PERFORMERS

To assess how posterior information is made accessible (in downstream readouts), we plot nonlinear probe loss against linear probe loss, evaluating the extent to which networks maintain a minimal probabilistic representation in a linear subspace (Figure 6). This analysis also assesses the degree to

which networks adhere to linear log-posterior codes, as suggested by PPC theory (Ma et al., 2006). These results reveal task-optimized networks do not consistently form weak probabilistic representations (Figure 6). In cue combination, linearization is trivial because both a probabilistic population code (PPC) and a copycat network (COPY) are able to construct the log-posterior linearly, a result guaranteed by using neural variability in the exponential family of distributions (Ma et al., 2006). Accordingly, the network also appears to maintain a linear code throughout training. More interestingly, in coordinate transformation, the copycat network is not able to linearly construct the log-posterior with the same fidelity as the PPC. Here, the task-optimized network develops a code that is more linear than the copycat network but not as linear as the PPC. Crucially, however, the final linear-nonlinear decoding gap (distance from unity) is not significantly different at the end of training (dark blue) relative to untrained networks (dark red). In Kalman filtering, we see little evidence of weak probabilistic representation: performers do not consistently learn to form a linear subspace in which the posterior is minimally represented. This crucially highlights the task-dependent nature of probabilistic representation: rather than being a generic property of task-optimization, the degree to which network form weak probabilistic representations depends heavily on the task. Some tasks may be trivially linear (like cue combination) whereas other tasks may support partial linearization (like coordinate transformation) and yet others offer no support for weak probabilistic representation (like Kalman filtering). Thus, in general, there is no clear or consistent evidence that task-optimized networks generically form weak probabilistic representations in nontrivial, complex tasks.

## 5 CONCLUSION

These results suggest that neural networks trained with non-probabilistic objectives do not generically develop probabilistic representations. In simple settings, they fail to form minimal codes that filter task-irrelevant information. Instead, they rely on input recoding strategies that linearize relevant aspects of target posterior distributions without explicitly compressing away information. We believe this framework will be pivotal not only in characterizing the structure of heuristic, non-probabilistic representations but also in identifying the exact inductive biases that promote probabilistic representation and generalization in artificial neural networks, such as width, loss function Lakshminarayanan et al. (2017), depth (Goodfellow et al., 2009), dropout (Srivastava et al., 2014; Achille & Soatto, 2017), and additive neural noise (Hinton & Van Camp, 1993). As this method is agnostic to network architecture, we also intend to apply this framework to convolutional neural networks and transformers to understand whether such architectures naturally induce probabilistic representation. Our framework would be particularly instrumental in clarifying debates about whether large language models implicitly learn world models (Li et al., 2024; Zhang et al., 2024).

Our framework is most applicable in domains with well-defined generative models, where representations can be evaluated against ground-truth posteriors. For extending our results to more complex tasks (e.g., image datasets), our approach would be applicable in settings well-approximated by generative models such as Gaussian Scale Mixtures (Orbán et al., 2016). Neural data pose additional challenges, as the structures of $\mathbf{X}$, $\mathbf{r}$, and $p_z$ are rarely known for certain; nonetheless, advances in latent-variable modeling will be crucial for extending our framework to biological recordings (Jensen et al., 2020; Schimel et al., 2021). Most importantly, task-optimal compression emerges as a hallmark of probabilistic computation, offering a principled way to distinguish probabilistic from non-probabilistic representations in biological networks and to reveal the structure of neural codes of uncertainty.

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

## A APPENDIX

### A.1 PROBE TRAINING DETAILS

All probes are trained on the hidden activations of the performer networks ($\mathbf{h}_{\text{perf}}$). For the static inference tasks, the nonlinear posterior and input probes are both two-layer ReLU networks. The linear posterior probe is a single layer network who output is passed through a softmax and compared to the Bayes-optimal posterior. The training objectives for each for the probes is as follows:

**Posterior probe**

$$\underset{\mathbf{W}_{\text{post}}, \mathbf{w}_{\text{post-readout}}}{\arg \min} \quad D_{\text{KL}}\left(p_{\text{true}}(z|\mathbf{x}_1, \mathbf{x}_2) || \underbrace{\text{Softmax}(\mathbf{w}_{\text{post-readout}}[\mathbf{W}_{\text{post}}\mathbf{h}_{\text{perf}}]_+)}_{\tilde{p}(z|\mathbf{x}_1, \mathbf{x}_2)}\right)$$

**Linear posterior probe**

$$\underset{\mathbf{w}_{\text{lin}}}{\arg \min} \quad D_{\text{KL}}\left(p_{\text{true}}(z|\mathbf{x}_1, \mathbf{x}_2) || \underbrace{\text{Softmax}(\mathbf{w}_{\text{lin}}\mathbf{h}_{\text{perf}})}_{\tilde{p}(z|\mathbf{x}_1, \mathbf{x}_2)}\right)$$

**Input probe**

$$\underset{\mathbf{W}_{\text{input}}, \mathbf{w}_{\text{input-readout}}}{\arg \min} \quad \mathcal{L}_{\text{NLL}}\left((\mathbf{x}_1\,\mathbf{x}_2), \underbrace{\text{Softplus}(\mathbf{w}_{\text{input-readout}}[\mathbf{W}_{\text{input}}\mathbf{h}_{\text{perf}}]_+)}_{=(\hat{\mathbf{x}}_1\,\hat{\mathbf{x}}_2)}\right)$$

For Kalman filtering, the nonlinear posterior and input probes had an additional ReLU layer (three in total). However, the nonlinear (and linear) posterior probe otherwise shared the same structure as those used for the static inference tasks. The input probe, however, was only trained to decode particular lags. Therefore, one input probe was trained for each lag shown in Figure 5. The input probes were trained with mean squared error loss on the true measurement of the appropriate lag $l$.

**Input (lag) probe**

$$\underset{\mathbf{W}_{\text{input}}, \mathbf{W}_{\text{hid-input}}, \mathbf{w}_{\text{input-readout}}}{\arg \min} \quad \mathcal{L}_{\text{MSE}}\left(x_{t-l}, \underbrace{\mathbf{w}_{\text{input-readout}}[\mathbf{W}_{\text{hid-input}}[\mathbf{W}_{\text{input}}\mathbf{h}_{\text{perf}}]_+]_+}_{=\hat{x}_{t-l}}\right)$$

## A.2 CONSTRUCTING THE PPC BENCHMARK

We have a log-posterior $\rho_z = \log p(z|\mathbf{X})$. PPC literature suggests that log-posteriors are a linear function of neural activity, *i.e.,*

$$\mathbf{A}\mathbf{r}_{\text{PPC}} + \mathbf{b} = \rho_z,$$

where $\mathbf{r}_{\text{PPC}}$ represents the hidden activations of the PPC, $\mathbf{A}$ is a matrix of tuning curves, and $\mathbf{b}$ is a bias term. Thus, to construct the PPC hidden layer, we wanted to find the hidden activations $\mathbf{r}$ corresponding to $\rho_z$. This is a simple least squares optimization problem that yields the well-known Moore-Penrose pseudoinverse:

$$\mathbf{r}_{\text{PPC}} \approx (\mathbf{A}^\top \mathbf{A})^{-1}\mathbf{A}^\top(\rho_z - \mathbf{b}).$$

For a well-conditioned and appropriately chosen (but otherwise generic) $\mathbf{A}$, this provides a valid representation of the optimal log-posterior. In practice, we assume that $\mathbf{A}$ consists of Gaussian tuning curves evaluated at a (sufficiently large) discrete set of latent stimulus values. This choice is natural because the posteriors in all tasks studied herein are Gaussian (or approximately so), and Gaussian tuning curves provide a robust basis for representing them. For a fair comparison across experiments, $\mathbf{r}_{\text{PPC}}$ was always constructed to match the dimensionality of the hidden layer of the task-optimized performer.

### A.3 COMPUTING THE 'MEAN ONLY,' FIXED VARIANCE PERFORMER BENCHMARK

We want to compare the information content of our performers to a non-probabilistic heuristic model that assumes *fixed* trial-to-trial variance but is able to perfectly decode the posterior mean (a 'mean-oracle'). This derivation assumes that the posteriors are Gaussian (or can be well-approximated with Gaussians). This is an appropriate assumption for our tasks, given that for the static inference tasks, we use Gaussian tuning curves and Poisson neural variability, which yield approximately Gaussian posteriors (Ma et al., 2006).

Assume that the true posterior on trial $i$ is $p_i(z) = \mathcal{N}(z|\mu_i, \sigma_i^2)$, and we want to approximate this with our fixed-width mean oracle $q_i(z) = \mathcal{N}(z|\mu_i, \hat{\sigma}^2)$. What should $\hat{\sigma}^2$ be?

$$\langle D_{KL}(p_i||q_i)\rangle_i = \frac{1}{2N}\left[\sum_i\left(\frac{\hat{\sigma}^2}{\sigma_i^2} + \ln\frac{\sigma_i^2}{\hat{\sigma}^2}\right) - N\right]$$

$$\frac{\partial}{\partial\hat{\sigma}^2}\langle D_{KL}(p_i||q_i)\rangle_i = \frac{1}{2N}\left(\sum_i\frac{1}{\sigma_i^2} - \frac{1}{\hat{\sigma}^2}\right)$$

$$0 = \sum_i\frac{1}{\sigma_i^2} - \frac{N}{\hat{\sigma}^2}$$

$$\frac{N}{\hat{\sigma}^2} = \sum_i\frac{1}{\sigma_i^2}$$

$$\hat{\sigma}^2 = \frac{N}{\sum_i\frac{1}{\sigma_i^2}}$$

### A.4 DERIVING THE MARGINAL KALMAN FILTERING POSTERIOR

Here, we take the standard Kalman filter, which is, importantly, conditioned on the parameters $\theta$ of the linear dynamical system, and marginalize out $\theta$. In our case, $\theta = \nu$, but this approach can be applied to any of the parameters in $\boldsymbol{\theta} = \{A, H, Q, \nu\}$.

$$p(z_{t+1}|x_{1:t+1}, \theta) = \int p(z_{t+1}, z_t|x_{t+1}, x_{1:t}, \theta)dz_t$$

$$\propto \int p(x_{t+1}|z_{t+1}, z_t, x_{1:t}, \theta)p(z_{t+1}|z_t, x_{1:t}, \theta)p(z_t|x_{1:t}, \theta)dz_t$$

$$= \int p(x_{t+1}|z_{t+1}, \theta)p(z_{t+1}|z_t, \theta)p(z_t|x_{1:t})dz_t$$

$$= p(x_{t+1}|z_{t+1}, \theta)\int p(z_{t+1}|z_t, \theta)p(z_t|x_{1:t})dz_t$$

$$= \pi(z_{t+1}|x_{1:t+1}, \theta)$$

$$p(x_{t+1}|x_{1:t}, \theta) = \int p(x_{t+1}, z_{t+1}|x_{1:t}, \theta)dz_{t+1}$$

$$= \int p(z_{t+1}|x_{1:t+1}, \theta)dz_{t+1}$$

$$= \bar{\pi}(x_{1:t+1}, \theta)$$

$$p(z_{t+1}|x_{1:t+1}, \theta) = \frac{\pi(z_{t+1}|x_{1:t+1}, \theta)}{\int \pi(z_{t+1} = z'|x_{1:t+1}, \theta)dz'}$$

$$= \frac{\pi(z_{t+1}|x_{1:t+1}, \theta)}{\bar{\pi}(x_{1:t+1}, \theta)}$$

$$p(x_{1:t+1}|\theta) = p(x_1, ..., x_t, x_{t+1}|\theta)$$
$$= p(x_1|\theta)p(x_2|x_1,\theta)p(x_3|x_2,x_1,\theta)...$$
$$= \prod_{\tau=1}^{t} p(x_{\tau+1}|x_{1:\tau},\theta)$$
$$= \prod_{\tau=1}^{t} \bar{\pi}(x_{1:\tau+1},\theta)$$
$$p(\theta|x_{1:t+1}) = \frac{p(x_{1:t+1}|\theta)p(\theta)}{\int p(x_{1:t+1}|\theta)p(\theta)d\theta}$$
$$= \frac{p(\theta)\prod_{\tau=1}^{t}\bar{\pi}(x_{1:\tau+1},\theta)}{\int \prod_{\tau=1}^{t}\bar{\pi}(x_{1:\tau+1},\theta)p(\theta)d\theta}$$
$$p(z_{t+1}|x_{1:t+1}) = \int p(z_{t+1}|x_{1:t+1},\theta)p(\theta|x_{1:t+1})d\theta$$

## A.5 SUPPLEMENTARY FIGURES

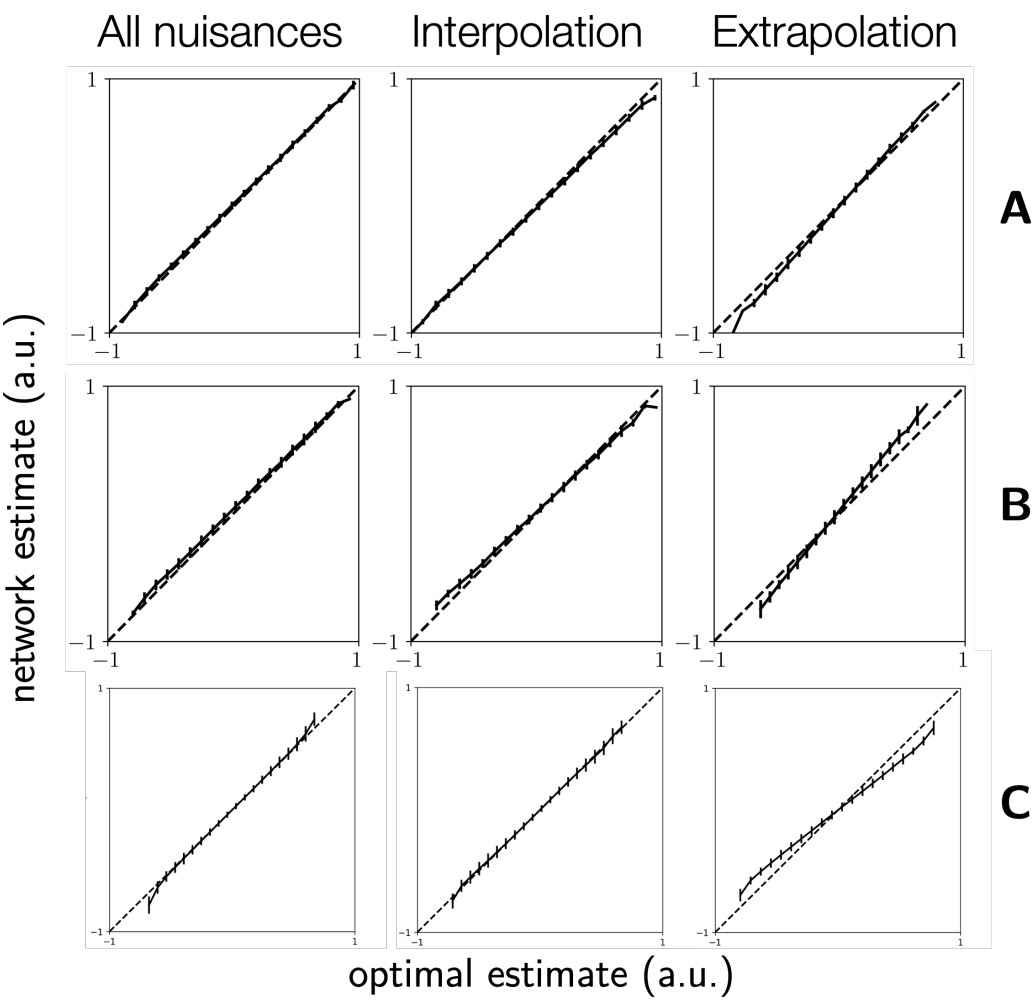

Figure 7: **Performers consistently behave in a Bayes-like manner, even under out-of-distribution nuisance generalization.** Performers output Bayes-optimal inferences in **A.** cue combination, **B.**, coordinate transformation, *and C.* Kalman filtering. For the two nuisance generalization conditions tested in Orhan & Ma (2017) ("all nuisances" and "interpolation"), performers are robustly Bayesian. Under a third generalization condition, "extrapolation," performance degrades mildly across all three tasks but still remains qualitatively similar to the Bayes-optimal estimates.

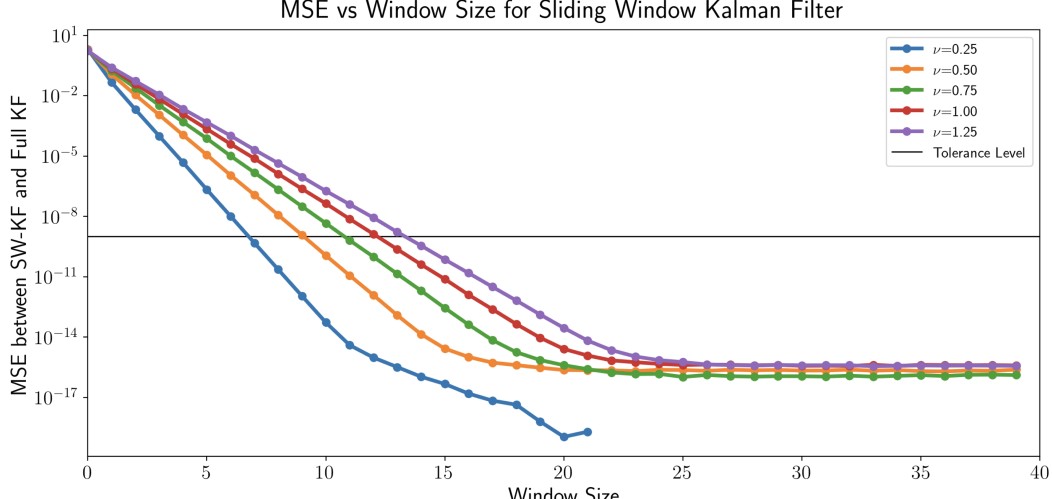

Figure 8: **Number of hidden neurons in the recurrent performer was chosen by comparing a sliding-window Kalman filter to a full Kalman filter.** For each nuisance parameter value $\nu$, we compared how close a sliding-window Kalman filter was to the full Kalman filter as a function of the window size for the sliding-window filter. Using a cutoff of $10^{-9}$, we selected a window size (or number of hidden recurrent neurons) of 13 for all performers.

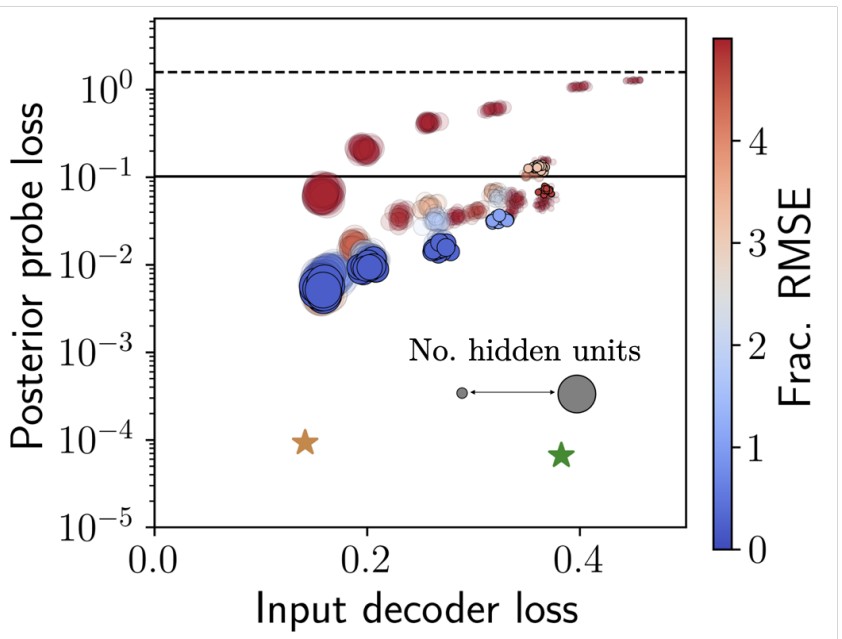

Figure 9: **Probabilisic representations do not emerge in networks performing cue combination under capacity constraints.** We test explicit lossy bottlenecks by training performer networks on cue combination (in the "all nuisance" generalization condition) as the number of hidden-layer neurons decreases ($n_{\mathrm{H}} \in \{200, 100, 50, 25, 12, 6\}$, labeled by icon size). If hidden layers could self-organize into probabilistic representations, decreasing the number of hidden neurons should have little impact on task performance (until the number of hidden neurons equals the number of sufficient statistics, in this case 4). However, we observe that network performance degrades considerably, corresponding with an increasing in *lossy* compression at the expense of uncertainty-awareness. Similar analysis was originally performed in (Orhan & Ma, 2017), though not using the fIB framework.

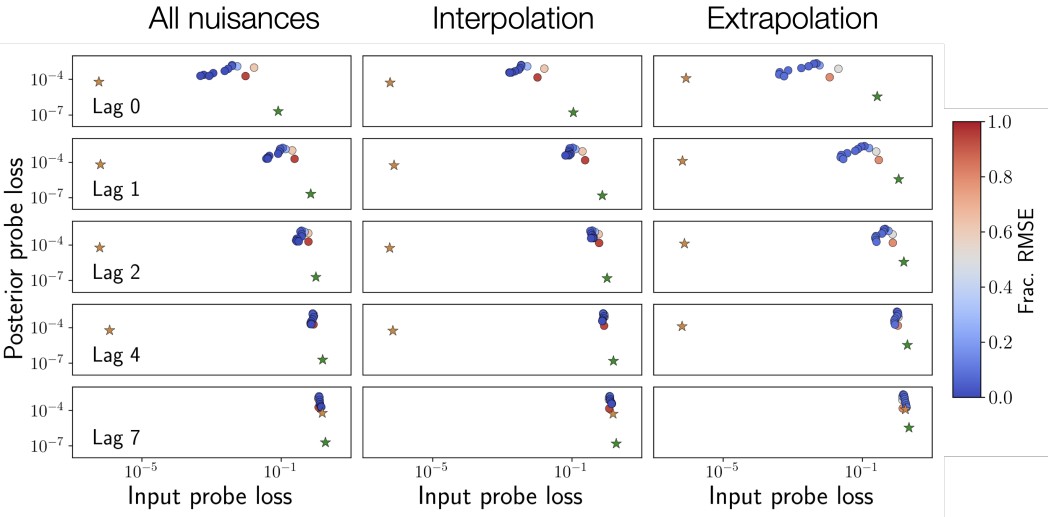

Figure 10: **Probabilistic representations fail to emerge in task-optimized RNNs even under strong resource constraints.** Consistent with Figure 5, RNNs do not optimally compress away input information even when the number of hidden neurons is smaller than the minimum window size $w$ to reconstruct the posterior from raw inputs. Networks also do not encode uncertainty with anymore fidelity than the copycat networks (orange stars), which are suboptimal by design.

