# OpenReview forum: "When sufficiency is insufficient: Probabilistic neural representations as an information bottleneck"
_ICLR.cc/2026/Conference — Submitted to ICLR 2026_

### Official Review · Reviewer_NLa8 · 2025-11-01

**Soundness:** 1
**Presentation:** 2
**Contribution:** 2
**Rating:** 2
**Confidence:** 5

**Summary:**

This paper proposes a test for emergent probabilistic representations in task-optimized neural networks, responding directly to Orhan and Ma 2017. They suggest that a probabilistic representation must be sufficient and minimal (invariant to nuisance variables). They then attempt to test whether simple networks optimized for simple tasks are indeed sufficient and minimal. They appeal to a "functional" information bottleneck concept, but instead of evaluating information, they use decoding approaches and compare decoding performance to two comparisons, a “copycat” network and a probabilistic population code (PPC), which should be, respectively, sufficient and minimal. They find that their trained networks are not minimal, and conclude that task-optimized networks do not represent probabilities.

**Strengths:**

The paper addresses an important question about what constitutes a probabilistic representation. They appeal to clear principles, and run appropriately simple tests. The probes are a reasonable way to approximate information content. Their results are interesting.

**Weaknesses:**

Giant problem:

They vastly overinterpret their results, and draw a much more dramatic conclusion than is warranted.
Their core premise is wrong: It is not a reasonable requirement that a probabilistic representation in the brain must be minimal. I'll give a few examples of why.

Let’s say a population encodes both p(x) AND log p(x), perhaps because the brain wants to marginalize sometimes and integrate cues sometimes. Or perhaps I encode some ancillary statistics in addition to the target distribution. These examples are both reasonable non-minimal representations of probability.

Their notion of minimality also breaks as soon as we consider pretty mundane population codes.

First, consider the whole brain. Then you could pretty clearly say that it is not a minimal representation of probability.

Second, consider only a subset of neurons, and evaluate this subset for their notion of sufficient and minimal. It seems their argument could conceivably work in such a localized code, treated in isolation.

Third, consider a multiplexed code. One subspace is encoding a probability distribution beautifully, but another orthogonal subspace in the same neurons is encoding something else. Then the authors' metric would say there is NOT a probabilistic representation because it’s not minimal, even though it's just a rotation of the second example of neuronal activity that were just deemed a probabilistic representation.

Because their premise is so problematic, it undermines the entire paper. If they dialed back their claim to be what they show — task-optimized networks do not compress optimally — then it would be a valid conclusion but not a strong one.




Medium problems:

In the Kalman filter, the copycat baseline was supposed to keep all inputs, but now they only use a subset in a sliding window. So it’s not copycat now. I don't understand how this is a useful comparison anymore. I can understand a sliding window truncation of the task-optimized filter, which should learn to compress. And the truncated Kalman filter is truncated at a long enough window so it has only a tiny error. But the copycat cannot be truncated this way without losing even approximate sufficiency.


Invariance to nuisances is directly related to properties of decoder. For example, if you use a square decoder, then it doesn’t matter that the code is invariant to sign.



Minor comments:

Figure 3: Where is r? Is this h? x?

Is this nu the noise VARIANCE? Above, Q is called process noise but it’s actually the process noise variance, I believe.

Figure 4 caption: I think they mean the prior is over the sum of uniformly distributed variables, p(z1+z2) which would be triangular, not that the distribution is a sum of distributions, which would not be triangular.

"Apparent parity with the PPC is misleading because of the log scale" — not sure that this is a significant difference at all. And if there IS parity with PPC, then it undermines their claims that their network is not minimal.

**Questions:**

Can you justify why minimality is a criterion for a probabilistic representation?

More concretely: Say I have an input i, a probability distribution q based on that input, and a localized, sufficient, minimal neural representation r_q of that probability distribution q. Let's then concatenate the input and r_q, and then rotate the combination (i, r_q) by some rotation matrix R to give a vector z = R.(i, r_q), then doesn't the population z have perfect information about the input, plus perfect information (that is perfectly formatted for a linear decoder) about the resultant probability? This is not minimal, correct? But would you agree that z is a representation of the probability?

---

> ### Author Response · Authors · 2025-11-26
> **Official Response by Authors to Reviewer NLa8 (1/2)**
>
> We sincerely thank the reviewer for their thoughtful and detailed feedback on our manuscript. The key concern raised by reviewer NLa8 is our use of **minimality** as a criterion for probabilistic representation. We appreciate the opportunity to clarify this point, as our intended claim is narrower and more defensible than the review presumes.
>
> We do **not** claim that any neural population that contains sufficient information to reconstruct a posterior must be minimal. Nor do we claim that the brain as a whole should exhibit minimality. Instead, our normative claim applies specifically to **representations used to _compute_ a posterior**, not to any population in which a posterior is merely _decodable_.
> In other words, minimality is a **functional signature** of probabilistic computation, not a universal property of probabilistic encoding.
> This distinction mirrors somewhat the taxonomy of Baker et al. (2021; updated unpublished 2025 manuscript via personal communication), which separates:
> 1. **information representations** (variables are decodable)
> 2. **usable representations** (variables appear in a computation-ready subspace)
> 3. **used representations** (variables are actually transformed in the computation)
>
> Minimality is only normative for levels (2) and (3); our manuscript explicitly investigates these levels, i.e., in the context that a relevant neural circuit is *explicitly used/usable* for probabilistic computation.
> We emphasize in the revision:
> - Our goal is not to prove that networks _cannot_ be probabilistic, but to provide a **principled test** for when probabilistic computation is **actually implemented** in neural representations. Using this, we show that trained networks do not generically meet these criteria.
> - Our framework separates:
>     - signals that merely _contain_ information about a posterior
>     - from signals that _use_ posterior-like quantities in computation (strong probabilistic representation)
>     - or at least facilitate easy computation with posterior-like quantities in downstream circuits (weak probabilistic representation)
>
> This distinction addresses the “readout fallacy:” a population may encode enough information to support a decoder even if that population is not performing Bayesian computation. Minimality (strong or weak) distinguishes these cases.
>
> Please see the corresponding edits (in red) made to the manuscript. We hope these provide adequate clarification to the existing ideas in the manuscript and adequately address the concerns of the reviewer. We look forward to the reviewer's feedback on these comments/revisions.

---

> > ### Author Response · Authors · 2025-11-26
> > **Official Response by Authors to Reviewer NLa8 (2/2)**
> >
> > > Can you justify why minimality is a criterion for a probabilistic representation?
> >
> > **Response:** If any neural circuit is performing Bayesian computation, then it must be throwing out some (task-irrelevant) information to perform that computation optimally. Therefore, minimality is a normative requirement only for neural populations that _perform_ probabilistic inference---not for all populations that _encode_ information relevant to inference.
> > Our argument follows Achille & Soatto (2016/2018) and Walker et al. (2023):
> > - A computation that transforms sensory variables $x$ into a posterior $p(z|x)$ must extract **sufficient statistics** of the posterior.
> > - Sufficient statistics should be **maximally invariant to nuisance variables**.
> > - Maximal invariance is equivalent to **minimality**.
> >
> > Thus, **if a neural population performs a probabilistic computation, its representation must be minimal with respect to irrelevant variability.** This does _not_ imply that the entire brain is minimal, nor that all populations encoding probabilistic quantities must be minimal. We now make this distinction explicit in the revision. Please see lines 151-159 (S2).
> >
> > > [...] Say I have an input i, a probability distribution q based on that input, and a localized, sufficient, minimal neural representation r_q of that probability distribution q. Let's then concatenate the input and r_q, and then rotate the combination (i, r_q) by some rotation matrix R...
> >
> > **Response:** We thank the reviewer for raising this point, as it highlights the strength and flexibility of our framework.
> > By our criteria, the reviewer is correct that $z=R(i,r_q)$ would not be a globally minimal probabilistic representation. However, the subspace corresponding to $r_q$ after rotation *is* minimal and sufficient. That is why we include two separate criteria for probabilistic representation. We clarify them here.
> > 1. **Strong minimality (or representational compression):** the representation globally compresses inputs into minimal sufficient statistics.
> > 2. **Weak minimality (or readout/subspace compression):** minimal sufficient statistics exist in a neural subspace that support downstream probabilistic computation.
> >
> > Therefore, under our criteria, $z$ *would* qualify as a **weak** probabilistic representation, i.e. one that doesn't necessarily *perform* probabilistic computation but facilitates downstream computation. We now make this distinction clear, using the reviewer's example explicitly. Please see new S2.1 and S2.2.
> >
> >
> > > Let’s say a population encodes both p(x) AND log p(x), perhaps because the brain wants to marginalize sometimes and integrate cues sometimes.
> >
> > **Response:** We agree that neural codes may be redundant and may support different kinds of computations. However, redundancy does not violate our criteria. If the reviewer means multiplexed linear codes (i.e. a DDC representation of $p(x)$ and a PPC representation of $\log p(x)$), then the code as a whole is still minimal with respect to $x$ even if the code itself is *redundant.*
> >
> > > First, consider the whole brain...
> >
> > **Response:** We strongly agree here, but our claim is not about the brain as a whole. We care only about **specific** functional circuits responsible for decision-making. We, therefore, now clarify that our criteria apply for "**specific** probabilistic representations" to avoid misinterpretation. Please see lines 151-159 (S2).
> >
> > > In the Kalman filter, the copycat baseline was supposed to keep all inputs, but now they only use a subset in a sliding window... the copycat cannot be truncated this way without losing even approximate sufficiency.
> >
> > **Response:** Our rationale is that the sliding-window copycat is not meant to represent a perfect buffer but a **matched-capacity upper bound**. The task-optimized RNN cannot store unbounded history, so a full copycat would give an unfair upper bound, since its memory capacity exceeds that of the RNN architecture. Therefore, a sliding-window copycat approximates the **best possible retention strategy under identical computational constraints**. Also, we see from the COPY (gold start, F5) performance that we do achieve approximate sufficiency using this matched-capacity copycat. We have revised the manuscript to state this more clearly. Please see S3.4 (lines 353-358).
> >
> > > Invariance to nuisances is directly related to properties of decoder. For example, if you use a square decoder, then it doesn’t matter that the code is invariant to sign.
> >
> > **Response:** We agree with the reviewer that probe structure interacts with measured invariances. To avoid overinterpretation, we now explicitly state that our decoders are intentionally high-capacity to minimize interaction between probe structure/performance. We plan to expand this analysis in future work. Please see lines 338-343 (S3.3).
> >
> > **Minor comments**
> > **Response:** We thank the reviewer for identifying these issues. All notation and figure explanations have been corrected accordingly.

---

### Official Review · Reviewer_mf7D · 2025-11-01

**Soundness:** 3
**Presentation:** 3
**Contribution:** 2
**Rating:** 6
**Confidence:** 2

**Summary:**

This paper addresses question in computational neuroscience: when can neural representations be considered genuinely probabilistic rather than heuristic recodings of input. The authors propose the functional Information Bottleneck (fIB) as a principled criterion for distinguishing minimal, sufficient probabilistic representations from non-minimal ones. They apply this framework across several canonical perceptual and inference tasks and challenge earlier conclusions claiming that probabilistic representations emerge naturally in task-trained neural networks.

**Strengths:**

The paper makes a clear and meaningful conceptual contribution by introducing the functional Information Bottleneck (fIB) as a more appropriate criterion for identifying when neural representations can be interpreted as genuinely probabilistic. The motivation is strong and well-articulated, and the empirical setups are carefully chosen to revisit influential claims in computational neuroscience. The comparisons against copycat and explicit probabilistic population code benchmarks provide a convincing frame of reference, and the takeaway—that probabilistic representations do not reliably emerge under standard task optimization—is important and likely to shift ongoing discussions in the field.

**Weaknesses:**

However, the empirical evidence is drawn from relatively simple tasks and network settings, which may limit how broadly the conclusions can be generalized to deeper or more complex architectures common in modern machine learning. Additionally, relying on minimality as a decisive criterion may be seen as a strong assumption—some representations might remain useful or functionally probabilistic without being globally minimal. Finally, the approach depends on probe-based evaluations, which can introduce sensitivity to decoder capacity and training dynamics, raising questions about how stable the conclusions are under different probing choices.

**Questions:**

I have a few questions regarding the paper: 1) How robust are the FIB conclusions to changes in probe capacity or architecture? Since the evaluation relies on decoder probes, it would be helpful to know whether the same outcomes hold when using weaker/stronger probes or alternative decoding setups. 2) Do the findings generalize to more complex, higher-dimensional models or tasks?
The current experiments focus on relatively small networks and controlled inference tasks. Testing fIB on deeper networks or tasks with richer structure could strengthen claims about the generality of the conclusions.

---

> ### Author Response · Authors · 2025-11-27
> **Official Response by Authors to Reviewer mf7D (1/2)**
>
> We sincerely thank the reviewer for their thoughtful feedback on our manuscript. We believe the reviewer's primary concern with our work is about the generality of our results. We appreciate the opportunity to clarify the scope of our claims and discuss our plans for future work beyond the scope of the present manuscript. We hope the clarifications below resolve some of the reviewer's concerns.
>
> > The empirical evidence is drawn from relatively simple tasks and network settings, which may limit how broadly the conclusions can be generalized to deeper or more complex architectures common in modern machine learning.
>
> **Response:** Our intention is not to generalize our results, i.e. to prove that networks _cannot_ be probabilistic but is instead to outline a robust framework for determining when a representation is probabilistic or not. Our claims also serve to refute the broad claim made in Orhan & Ma that end-to-end optimization necessarily encourages probabilistic representation, which we demonstrate is not true even in simple probabilistic tasks. Indeed, the tasks and networks were intentionally made simple to demonstrate our framework in settings that permit tractable (analytical) posteriors and unambiguous network analysis. While we intend to study more complicated tasks (image datasets, etc.) and different architectures (deeper MLPs, transformers) in future work, we believe the suite of simple probabilistic tasks, which are canonical in experimental neuroscience and were studied previously in the context of probabilistic representations, provides a comprehensive demonstration of our fIB framework. That said, we have performed analysis of deeper networks performing cue combination; we would be happy to include some of those results in the Appendix if the reviewer felt it would strengthen our claims.
> We have updated our manuscript to more clearly discuss the rationale for simple settings. Please see linear 89-96.
>
> > Relying on minimality as a decisive criterion may be seen as a strong assumption—some representations might remain useful or functionally probabilistic without being globally minimal.
>
> **Response:** We thank the reviewer for this point as it provides an opportunity for us to clarify the hierarchy of our proposal (which we now make more explicit in our revised manuscript). The primary minimality criterion, which we refer to as representational compression (deleting input information), was our test for "strong" probabilistic representation. This means that a particular representation is itself *performing* a probabilistic computation so that downstream circuits are only able to use posterior information and nothing more. Our linear decodability results (S4.2) served as a softer or "weak" probabilistic representation test. Here, we are implicitly arguing not that a probabilistic representation must be globally compressed but that at least in some linear subspace, we have a compressed sufficient representation. This aligns with the reviewer's point, and we have made this hierarchy (strong vs. weak representation) explicit in S2.1 and S2.2 of the revised manuscript.
>
> > The approach depends on probe-based evaluations, which can introduce sensitivity to decoder capacity and training dynamics, raising questions about how stable the conclusions are under different probing choices.
> > [...] How robust are the FIB conclusions to changes in probe capacity or architecture? Since the evaluation relies on decoder probes, it would be helpful to know whether the same outcomes hold when using weaker/stronger probes or alternative decoding setups.
>
> **Response:** We appreciate this concern very much. We intentionally designed nonlinear probing (i.e. to test strong probabilistic representation) as high capacity is possible (within reason) to ensure that if information was decodable given enough data, we would be able to detect it. However, for linear probing analysis and even for the exact choices of nonlinear probes, we agree that there are several alternatives we should test in the future. Indeed, a useful alternative to the linear probe would be CCA or variants of regularized regression, and we believe it would be fruitful to examine the degree of capacity need to decode some of these quantities: if only high-capacity decoders can decode inputs, then it may be more relevant to consider generic (lower-capacity) neural circuits as being invariant to those nuisances. We have added a brief discussion of these ideas in S3.3 (lines 338-343).

---

> > ### Author Response · Authors · 2025-11-27
> > **Official Response by Authors to Reviewer mf7D (2/2)**
> >
> > > Do the findings generalize to more complex, higher-dimensional models or tasks?
> >
> > **Response:** We agree with the reviewer that strengthening our claims about the fIB hinges on applying it to more complex datasets. We have tested the results on a variant of Kalman Filtering (Dayan, Kakade, Montague, 2000) where the posterior uncertainty dynamics are modulated directly by the input (rather than being governed by simple monotonic dynamics to a steady-state). Even in this task, where posterior uncertainty is even more important to perform optimally, we see similar results to the standard KF. We would be happy to add these results to the Appendix if the author felt it would strengthen our claims.
> >
> > As the reviewer points out, though, all of these are still relatively low-dimensional tasks. We are currently extending this to higher dimensional filtering tasks as well as HMM filtering and POMDPs (where tracking belief states follows a similar logic). We are also interested in applying this to image-computable generative models like the Gaussian Scale Mixture model, which would allow us to directly apply our framework to realistic experimental stimuli (such as oriented gratings). A brief discussion of this is now included in the conclusion (lines 519-523).

---

### Official Review · Reviewer_79Dd · 2025-11-01

**Soundness:** 3
**Presentation:** 3
**Contribution:** 4
**Rating:** 8
**Confidence:** 4

**Summary:**

The paper argues that neural networks trained on standard inference tasks don’t actually form true probabilistic representations of uncertainty. The authors introduce a framework called the functional Information Bottleneck (fIB), which tests whether hidden-layer representations are both sufficient (contain all task-relevant information) and minimal (contain nothing extra, especially related to inputs).
Using this method, they find that networks trained on cue combination, coordinate transformation, and Kalman filtering perform well but only recode inputs heuristically rather than compressing them into genuine Bayesian posteriors.

**Strengths:**

1/ Conceptual Clarity and Theoretical Innovation. The notion that probabilistic representation = sufficiency + minimality is elegant and clarifies an ambiguity in neuroscience and AI. Additionally, the fIB framework is a tractable, practical adaptation of the classic Information Bottleneck, avoiding the problems of estimating mutual information.

2/ Methodological Rigor. Thorough experimental design, with controls such as “copycat” and “probabilistic population code” benchmarks, strengthens the conclusions.

3/ Challenges the existing literature. The paper challenges a popular but weakly supported belief that probabilistic computation “emerges naturally” in neural nets. By showing the absence of minimality, it redefines what counts as genuine probabilistic representation.

**Weaknesses:**

1/ Limited empirical scope. The experiments focus on simple, low-dimensional tasks (cue combination, coordinate transformation, 1D Kalman filtering). It’s unclear whether the conclusions hold for more complex, high-dimensional networks or real sensory data.

2/ Dependence on probe networks. The fIB results depend on the behavior of trained probes, whose success depends on architecture and capacity. This makes it difficult to tell whether poor decoding reflects genuine information loss or simply probe limitations.

3/ No recovery (positive) case shown. The paper convincingly shows what doesn’t produce probabilistic representations but lacks an example where such representations do emerge. Without a successful counterexample, the framework’s diagnostic power remains one-sided.

**Questions:**

1/ Could minimal probabilistic representations emerge naturally if networks were trained with architectural bottlenecks, capacity limits, or noise regularization (e.g., dropout or variational objectives)?

2/ Recovery Cases. Related to the first point, what conditions or training regimes might allow a network to pass the fIB test — i.e., to develop representations that are both sufficient and minimal? Could explicitly probabilistic objectives or generative models serve as such recovery cases?

3/ Question about linear input probes. How sensitive are the fIB results to the choice and capacity of probe networks? Would defining minimality using linear rather than nonlinear input probes lead to different conclusions about what information is “functionally removed”?

4/ Broader Implications: If networks (and perhaps biological systems) can behave Bayesianly without encoding full probabilistic posteriors, what does that imply for how we interpret neural data and define “probabilistic computation” in the brain?

---

### Official Review · Reviewer_hD26 · 2025-11-04

**Soundness:** 2
**Presentation:** 1
**Contribution:** 2
**Rating:** 2
**Confidence:** 3

**Summary:**

The paper proposes a quantitative criterion for qualifying internal representations of neural networks as "probabilistic" or "heuristic". For this purpose, internal representations in simple DNNs that perform cue integration, coordinate transformation and temporal filtering tasks are analyzed by "probe networks" that analyze whether the internal representation can be used to solve the problem, and/or to reconstruct the input. It concludes that the investigated DNNs do not form probabilistic internal representations without explicit additions to the loss function.

**Strengths:**

The paper tackles a long-standing problem, namely the encoding of information in biological and artificial NNs. It proposes a quantitative criterion to qualify an internal representation as "probabilistic" that could, in principle, be applied to any DNN. The experiments are well-described and the results support the claims that are made.

**Weaknesses:**

- The paper is easy to read but the message of the paper is not easy to understand. It is not made very clear what conceptual advantages a "probabilistic" internal representation could have, and whether it is a generally desirable thing to achieve in DNNs
- Likewise, it is not clear what the applicability of these results is. Like, would it improve classification results on MNIST if internal representations were probabilistic? Probably not, but what kinds of problem *could* be solved in this case is not made clear.
- The experiments are extremely simplistic, and so are the DNNs used.
- The results, on very simple DNNs,  are essentially only negative, which is fine, but a wider investigation about how to *force* DNNs to develop probabilistic internal representations would complement this very well.

**Questions:**

- Please explain why the used networks and problems are so simple. Or rather: could all of this be applied to more complex problems, like, e.g., image classification? Or speech recognition? It seems that especially cue integration happens there as well, so this might be a far better motivation for this type of research
- Can the method described here be applied to any DNN? E.g., a LLM or ResNet-18?

---

> ### Author Response · Authors · 2025-11-26
> **Official Response by Authors to Reviewer hD26 (1/2)**
>
> We are grateful to the reviewer for their thoughtful critique of our manuscript and sincerely thank them for their constructive feedback. We believe that the key concern raised by reviewer hD26 reflects a misunderstanding about the intended scope of our contributions. Our work is grounded primarily in computational and theoretical neuroscience, where interpreting neural representations through the lens of probabilistic generative models is both standard and essential. In contrast, many classical machine learning benchmarks (e.g., MNIST classification or large language models) do not rely on explicit generative models and therefore are not well suited for evaluating Bayesian representational criteria of the kind we present herein.
> We appreciate the opportunity to clarify the neuroscientific context in which our framework makes strong and novel contributions, and to describe more precisely the conditions under which our approach can also inform machine-learning settings. We hope that the explanations below resolve the reviewer’s concerns by making these domain differences explicit and by highlighting the conceptual and methodological contributions of our work. Thank you!
>
> All relevant revisions have been added to an updated version of the manuscript (in red text).

---

> ### Author Response · Authors · 2025-11-26
> **Official Response by Authors to Reviewer hD26 (2/2)**
>
> > It is not made very clear what conceptual advantages a "probabilistic" internal representation could have, and whether it is a generally desirable thing to achieve in DNNs
>
> **Response:** We appreciate that our manuscript did not sufficiently articulate the functional benefits of probabilistic representation. We have revised the introduction to summarize the points made in Koblinger et al., 2021, which clarify that “probabilistic representation” enable 1) modularity and reuse across computational circuits, 2) robustness to information fusion under nuisance variation, 3) active sensing, and 4) learning.
> We also note that probabilistic representation is closely related to adversarial robustness, in that a representation that is invariant to nuisance perturbations is explicitly one that is robust to adversarial samples. We hope this clarifies both the neuroscience and machine learning motivations behind our work.
> Please see lines 42-48 (intro).
>
> > Likewise, it is not clear what the applicability of these results is. Like, would it improve classification results on MNIST if internal representations were probabilistic?
>
> **Response:** Our current approach is not applicable to classification tasks on MNIST only because MNIST does not have a well-defined generative model. Our primary contribution in this work was to develop a simple theory for identifying probabilistic representations in (largely simple) tasks where the generative model permits analytical solutions. However, we do intend to apply these to more real-world settings such as MNIST, where we can define an explicit generative model (like a Gaussian Scale Mixture model) and determine whether networks that develop probabilistic representations are more adversarially robust than networks that are purely discriminative classifiers. We have included a brief discussion of this point in our conclusion.
> Please see lines 525-531.
>
> > The experiments are extremely simplistic, and so are the DNNs used.
>
> **Response:** Our networks and tasks are intentionally simplistic. These tasks are the canonical tasks for probabilistic computation in neuroscience (e.g., cue integration, Kalman filtering, coordinate transformation). They are precisely the settings where the relevant posterior is analytically known and experimental evidence suggests that humans perform probabilistic computation, allowing us to test whether such representations are truly probabilistic. In fact, using more complex datasets (e.g., images) would obscure the underlying computational question because the ground-truth posterior is unknown and one could not meaningfully validate our fIB framework. Finally, our work was directly inspired as a response to previous work (like Orhan & Ma, 2017) that use the same (or very similar) tasks on networks with very similar architectures. We have updated the manuscript to clearly state these justifications.
> Please see lines 89-96 (intro).
>
> > The results, on very simple DNNs, are essentially only negative, which is fine, but a wider investigation about how to _force_ DNNs to develop probabilistic internal representations would complement this very well.
>
> **Response:** The central premise of our work is precisely that even in simple probabilistic tasks, networks do not always form probabilistic internal representations. This is an important negative result because it demonstrates that probabilistic representation does not emerge naturally from task optimization. However, we agree with the reviewer that exploring methods for enforcing probabilistic representation would be an exciting future direction, as we discuss briefly in our Conclusion. We have tested this framework on networks of increasing depth and neural noise, but these inductive biases still do not encourage probabilistic representation generically. We have also tested a second variant of the Kalman Filter (dubbed the conditional Kalman Filter, (Dayan, Kakade, Montague 2000)), where the posterior uncertainty (and therefore Kalman gain) is input-dependent. This allows for richer posterior dynamics and, therefore, an increased task-dependent incentive to represent uncertainty. However, we see very similar results and no strong signs of probabilistic representation even in this task. We would be happy to include these results in the Appendix if the reviewer believes it would strengthen our claims.
>
> > Please explain why the used networks and problems are so simple...
>
> Please see our previous responses.
>
> > Can the method described here be applied to any DNN?
>
> Yes, our framework is relatively architecture-agnostic. We intend to apply our framework to transformer architectures in particular, as there has been fruitful discussion about the extent to which LLMs build implicit world models that reflect the underlying statistics of the world. We believe our framework would be instrumental in clarifying such claims. We have edited the manuscript to include these discussions.
> Please see lines 520-524 (conclusion).

---

### Meta-Review · Area_Chair_GCg1 · 2026-01-07

**Summary:**

This paper proposes the functional Information Bottleneck (fIB) as a criterion for identifying probabilistic neural representations. The reviewers agree that the paper addresses an important question and the motivation is clear. The reviewers' common concerns lie in overinterpretation of the results, strong conceptual assumptions, limited empirical scope, and heavy dependence on probe-based evaluations. In addition, the authors did not respond to any of the questions and concerns raised by Reviewer 79Dd. Overall, I do not believe this submission meets the standard for acceptance.

**Reviewer Concerns:**

After the rebuttal, some concerns were partially addressed. The authors clarified that the use of simple tasks is intentional and necessary for analytical tractability, acknowledged the absence of recovery cases, and narrowed the stated scope of their claims.

However, several concerns remain unresolved. As for the validity of the minimality criterion, Reviewer NLa8 raises that "It is not a reasonable requirement that a probabilistic representation in the brain must be minimal." Additionally, the framework’s reliance on probe networks continues to raise questions about sensitivity to decoder capacity and training dynamics. Concerns about limited empirical scope and lack of evidence on more complex tasks also remain.

**Reviewer Scores:**

The rebuttal does not fundamentally change the reviewers’ assessments or address their core concerns. Moreover, the absence of any response to Reviewer 79Dd further weakens the submission. Thus, I do not expect reviewer scores to increase.

---

### Decision · Program_Chairs · 2026-01-26

Reject